# Luteolin Induces Selective Cell Death of Human Pluripotent Stem Cells

**DOI:** 10.3390/biomedicines8110453

**Published:** 2020-10-27

**Authors:** Young-Hyun Go, Jumee Kim, Ho-Chang Jeong, Seong-Min Kim, Yun-Jeong Kim, Soon-Jung Park, Sung-Hwan Moon, Hyuk-Jin Cha

**Affiliations:** 1Department of Life Science, Sogang University, Seoul 04107, Korea; younghyun1992@gmail.com (Y.-H.G.); hcjeong72@gmail.com (H.-C.J.); 2College of Pharmacy, Seoul National University, Seoul 08826, Korea; jumeekim11@gmail.com (J.K.); seongminkim96@gmail.com (S.-M.K.); yunjkim36@gmail.com (Y.-J.K.); 3Stem Cell Research Institute, T&R Biofab Co., Ltd., Siheung 15073, Korea; sjpark@tnrbiofab.com (S.-J.P.); sunghwanmoon@tnrbiofab.com (S.-H.M.); 4Research Institute of Pharmaceutical Sciences, Seoul National University, Seoul 08826, Korea

**Keywords:** flavonoid, quercetin, teratoma, luteolin, human pluripotent stem cells

## Abstract

Despite recent advances in clinical stem cell therapy applications based on human pluripotent stem cells (hPSCs), potential teratoma formation due to the presence of residual undifferentiated hPSCs remains a serious risk factor that challenges widespread clinical application. To overcome this risk, a variety of approaches have been developed to eliminate the remaining undifferentiated hPSCs via selective cell death induction. Our study seeks to identify natural flavonoids that are more potent than quercetin (QC), to selectively induce hPSC death. Upon screening in-house flavonoids, luteolin (LUT) is found to be more potent than QC to eliminate hPSCs in a p53-dependent manner, but not hPSC-derived smooth muscle cells or perivascular progenitor cells. Particularly, treating human embryonic stem cell (hESC)-derived cardiomyocytes with LUT efficiently eliminates the residual hESCs and only results in marginal effects on cardiomyocyte (CM) functions, as determined by calcium influx. Considering the technical limitations of isolating CMs due to a lack of exclusive surface markers at the end of differentiation, LUT treatment is a promising approach to minimize teratoma formation risk.

## 1. Introduction

Human pluripotent stem cells (hPSCs) (e.g., human embryonic stem cells (hESCs) and human induced pluripotent stem cells (hiPSCs)) are considered promising stem cell sources for future regenerative medicine due to their pluripotency, which enables the production of any type of somatic cell [1]. The recent promising outcome in the clinical trials with retinal pigment epithelial (RPE) cell [2] or dopamine progenitor cell [3] from hPSCs, demonstrates the regenerative potential of stem cell therapy. However, for a wide range of clinical applications, the potential risks of hPSC-based cell therapy should be first resolved to ensure their safety [4]. One of the risk factors of this technology is the potential for on-site teratoma formation derived from the active proliferation potential of residual undifferentiated hPSCs, which remains a serious technical challenge for the safe application of hPSC-based stem cell therapy [5,6]. Therefore, a variety of strategies have been developed to lower these risks [7].

Particularly, small molecule treatment to induce selective undifferentiated hPSC death (i.e., ‘stemotoxic’ agents) immediately prior to engraftment is a preferred strategy due to its simplicity when it ensures the functional safety of differentiated cells [8]. Therefore, a variety of small molecules have been identified/developed to selectively eliminate hPSCs and inhibit teratoma formation with minimal effects on differentiated cells [9,10,11,12]. Undifferentiated hPSCs, for example, show a high vulnerability to YM155 [9], which was initially developed to inhibit survivin [13] while leaving hPSC-derived human mesenchymal stem cells functionally intact [14] via the selective cellular uptake of YM155 due to high solute carrier SLC35F2 protein expression in hPSCs [15].

Similarly, quercetin (QC; i.e., a natural flavonoid with anti-oxidant properties) also was identified as a stemotoxic flavonoid [9] whose selective effect on hPSCs results from mitochondrial permeability transition (MPT) [16]. QC is widely used as a dietary supplement and its administration is safe to normal somatic cells [17,18]. Therefore, given the safety of QC toward hPSC-derived dopaminergic neurons [9], the compound was used to eliminate undesirable undifferentiated hiPSCs in dopaminergic progenitor cell clinical trials in Parkinson’s disease patients [3]. Considering the no stemotoxic effect of both kaempferol (KP), which shares close structural similarities with QC [9], and YM155 derivatives lacking nitrogen in their pyrazine structure [19], the structure/activity relationship of these flavonoids was expected to result in ‘stemotoxic’ effects.

Flavonoids possess a common phenylchromen structure (C6: A ring -C3: C ring -C6: B ring) with one or more hydroxyl groups and are categorized into flavonols (3-hydroxyl-2-phenylchromen-4-one; e.g., myricetin (MC), quercetin (QC), and kaempferol (KP)) and flavones (2-phenylchromen-4-one; e.g., chrysin (CHY), apigenin (API), and luteolin (LUT)) depending on the occurrence of a 3-hydroxyl group in the central pyran-4-one ring. A recent study demonstrated that the anticancer effect of flavones differs from that of flavonols [20], suggesting that the number of hydroxyl groups in flavonoid structures may determine their cytotoxic effects on cancer cells or hPSCs.

During this study, we screen a total of 6 in-house flavonoids with different hydroxyl groups to identify compounds with potent stemotoxic effects and find that LUT, a flavone with two hydroxyl groups at the B ring (i.e., similar to QC), possess more potent stemotoxic effects than QC. No hPSC-derived differentiated cell death occurs and hPSC-derived cardiomyocyte calcium influx remains unaltered, highlighting the high selectivity of LUT toward undifferentiated hPSCs.

## 2. Materials and Methods

### 2.1. Cell Culture and Reagents

Human embryonic stem cell (WA09: H9, WiCell Research Institute, Madison, WI, USA) and human induced pluripotent stem cell lines [21,22] were maintained in iPSC-brew (Miltenyi biotechnology, #130-104-368) with 0.1% gentamycin (Gibco, Waltham, MA, USA, #15750-060) on a matrigel (Corning, Corning, NY, USA, #354277)-coated cell culture dish at 37 °C and humidified to 5% in a CO_2_ incubator. Quercetin (#Q4951), luteolin (#L9283), kaempferol (#11852), chrysin (#C80105), apigenin (#10798), and the glycoside analogues were purchased from Sigma–Aldrich and kindly provided by Prof. Sang–Hyun Sung (College of Pharmacy, Seoul National University). The human perivascular progenitor cells (PVPCs) derived from hESCs [23] (kindly gifted from Dr. Moon Sung–Hwan, T&R Biofab Co., Siheung, Korea) and human smooth muscle cell derived from human iPSCs [21] (kindly gifted from Dr. Tae–Hee Lee, Sejeong University) were maintained in EBM2 medium (Lonza, #CC-3156) with 0.1% gentamycin at 37 °C humidified to 5% in a CO_2_ incubator. Concerning cardiomyocyte differentiation and culture, RPMI-1640 (#R8758) and advanced-MEM (Gibco, #12492013) were used.

### 2.2. Cardiomyocyte Differentiation

Cardiomyocyte differentiation from hESCs was performed as previously described [24]. Briefly, cells were seeded onto a hPSC qualified Matrigel (Corning, #354277)-coated cell culture dish at 140,000 cells/cm^2^ dish. Five micrometers of Y-27632 (Peprotech, #1293823) was added for the first 24 hours after passage. The medium was changed daily, and cells were allowed to grow in StemMACS^TM^ iPSC-brew (Miltenyi Biotec, Bergisch Gladbach, Germany, #130-1040368) for 3–4 days until the cells were 90% confluent. Occurring at day 0, cells pretreated with CHIR99021 (Peprotech, Rocky Hill, NJ, USA, #2520691), were cultured with 8 μM/mL in CDM (cardiomyocyte differentiation medium: RPMI1640 (Sigma–Aldrich, St. Louis, MO, USA, #R8758)/BSA (Sigma–Aldrich)/ascorbic acid (Sigma–Aldrich, #A92902)). After 48 hours, the medium was changed to CDM supplemented with 3 μM/mL C59 (Wnt inhibitor, #1248913, Peprotech, NJ, USA) for an additional 48 h. Occurring on day 5, the medium was replaced with CDM and changed with fresh every 2 days. Spontaneously, contracting cells began to appear at day 6 to day 10. Between day 10 to day 15, the medium was replaced with CDM containing L-lactic acid to metabolically select and purify cardiomyocytes (hESC-CMs). All live images were taken with a Lumascope microscope (Etaluma, Carlsbad, CA, USA).

### 2.3. Cell Death Analysis

Cell death was analyzed by flow-cytometry as described previously [25]. Regarding Annexin V/7-AAD staining, cells at 24 h after treatment of each flavonoid were washed twice with PBS and stained with FITC conjugated Annexin V antibody (BD Bioscience, Franklin Lakes, NJ, USA, #556419) and 7-AAD (BD Bioscience, #559925) for an additional 45–60 min at room temperature in the dark. Cells stained with Annexin V/7-AAD were analyzed by FACS Calibur or FACS Lyric (BD Bioscience). Concerning all of the bright field images captured, a Light channel optical microscope (Olympus, Tokyo, Japan, CKX-41) or JULI-stage (NanoEntek, Seoul, Korea) was used in accordance with the manufacture’s protocol. The activity of caspase-3 was analyzed using a colorimetric active caspase-3 assay kit (Sigma–Aldrich, #CASP3C) according to the manufacture’s protocol.

### 2.4. Fluorescence-Based Competition Assay

A fluorescence-based cell death competition assay was performed as described previously with slight modifications [14]. The same number of green fluorescent protein (GFP)-expressing hESCs and PVPCs were co-cultured under hESC culture conditions. Cells were live-monitored with the JULI-stage, fluorescence live image system (NanoEntek, Seoul, Korea) or by flow-cytometry analysis using FACSCalibur (BD Biosciences) after treatment of LUT.

### 2.5. Immunofluorescence and Immunoblotting

Immunoblotting and immunofluorescence assays were performed as described previously [26]. Antibodies for PARP-1 (#sc-8007) and α-tubulin (#sc-8035) were purchased from Santa Cruz. Cleaved caspase-3 (#9661S), and cleaved caspase-9 (#9505S) were purchased from Cell Signaling Biotechnology. Counter staining of the nucleus was performed by 4′,6-Diamidino-2-phenylindole dihydrochloride (DAPI) (Sigma–Aldrich, #D9542).

### 2.6. RNA Extraction and Quantitative Real-Time PCR Assay

Total RNA was isolated with the easy-blue total RNA extraction kit (Intron, #17061, Seongnam, Korea). 1 µg of total RNA was converted to cDNA with the Prime Script RT Master Mix (Takara, #RR036A, Kusatsu, Japan) in accordance with the manufacturer’s protocol. Quantitative real-time PCR analysis was performed using TB green premix taq (Takara, #RR820A, Kusatsu, Japan) on a LightCycler 480 II (Roche, Basel, Switzerland) in accordance with the manufacturer’s protocol. Primer information follows (Table 1).

### 2.7. Flow-Cytometry

Cells were fixed with fix-permeabilization solution (BD Bioscience, #554722) for 10 min at room temperature or 1 h at 4 °C. The fixed cells were washed with PBS twice and then were blocked with 5% bovine serum albumin (Biosesang, Seongnam, Korea, #SR2195-050-00) solution in TBS buffer. After the blocking, cells were washed with TBS twice and incubated with CD172α/β (BioLegend, San Diego, CA, USA, #323808) with the fluorescent (PE-Cy7) conjugated antibody in accordance with the manufacturer’s concentration offered. Stained cells were analyzed by FACS Calibur or FACS Lyric (BD Bioscience) according to the manufacturer’s protocol.

### 2.8. Measurement of Intracellular Ca^2+^ Influx

Regarding the measurement of calcium oscillation in human embryonic stem cell-derived cardiomyocyte, cells were cultured with 2 µM of Flou-4, AM (Invitrogen, Carlsbad, CA, USA, #F14201) for 40 min in a 37 °C humidified incubator. After staining of Flou-4, AM, cells were quickly washed twice by advanced-MEM and filled with CM culture media. A Lumascope (Etaluma, Carlsbad, CA, USA) and the software Lumaview (Etaluma, Carlsbad, CA, USA) were used for recording and capturing images of a bright field or green fluorescent protein GFP channel. To analyze the captured images, LumaQuant8 (Etaluma, Carlsbad, CA, USA) software were used in accordance with the manufacturer’s calcium oscillation protocol.

### 2.9. Statistical Analysis

The graphical and quantification data were presented as mean ± S.D. Statistical significance among the three groups and between groups was determined using one-way analysis of variance (ANOVA) following a Tukey Post-test and a Student’s *t*-test, respectively. Statistical analysis was performed with GraphPad Prism 8 software (https://www.graphpad.com/scientific-software/prism/). Significance was assumed for *p* < 0.05 (*), *p* < 0.01 (**), *p* < 0.001 (***), *p* < 0.0001 (****).

## 3. Results

### 3.1. Stemotoxic Screening of Flavonoids

As observed in many types of cancer cells, the ATP production of hPSCs relies on glycolysis rather than oxidative phosphorylation (OXPHOS), even in the presence of high levels of oxygen [27]. Therefore, hPSCs express higher levels of *SLC2A1* (Solute Carrier Family 2 Member 1), encoding GLUT-1, a glucose transporter protein than human dermal fibroblasts (hDFs) (Appendix A). As the conjugation of glucose to drug molecules has been widely applied to improve the delivery of drugs to brains [28,29] or cancers [30,31] with high expression of glucose transporters, quercetin glycoside (QC-GLU) may have more potent stemotoxic properties against undifferentiated hPSCs than QC. To examine this possibility, we determined the stemotoxic effects of both QC and QC-glycoside (i.e., QC-7-O-glycoside). Unexpectedly, QC glycoside exhibited no stemotoxic effects as similar other glycosides (Appendix A). We had previously demonstrated the non-stemotoxic effects of kaempferol (KP), which shares a similar chemical structure with QC (two hydroxyl groups in QC vs. one hydroxyl group in KP at the B ring) [9], suggesting that other flavonoids with different numbers of hydroxyl groups may have more potent stemotoxic effects on hPSCs. To test this, in-house flavonoids with different numbers of hydroxyl groups at the C and B rings were screened (Figure 1A). A total of six in-house flavonoids were classified as flavones or flavonols depending on the presence of hydroxyl groups at R1 in the C ring (Figure 1A,B). The initial screening of the effect of the six flavonoids on undifferentiated hESCs was performed with a single dose to broadly characterize their stemotoxic effects (i.e., induction of cell death in hESCs). Shown in Figure 1B, hESCs manifested alterations in cell morphology 24 h after QC or luteolin (LUT) treatment. KP, myricetin (MYC), and chrysin (CHY) exhibited negligible stemotoxic effects, suggesting that the hydroxyl group in R1 of the C ring and the number of hydroxyl groups in the B ring determine the degree of a compound’s stemotoxic effects. To quantify stemotoxic effects more precisely, cell death was quantified via flowcytometry. Consistent with the results in Figure 1B, hESC death was evident after apigenin (API), luteolin (LUT), and QC treatment (Figure 1C). Moreover, it is worth noting that MYC, which possesses three hydroxyl groups in the B ring (i.e., one more hydroxyl group than QC), showed only minimal effects. Similar to QC glycoside, glycoside of KP and LUT had a negligible effect on hPSCs (Appendix A).

### 3.2. Potent Stemotoxic Effects of Luteolin

After determining the stemotoxic effect of flavonoids with a single dose (50 µM) (Figure 1B,C), three flavonoids (API, LUT and QC) were selected for further validation to examine the dose-dependent effects toward undifferentiated hESCs. These experiments revealed that LUT was considerably more potent than the other flavonoids. LUT treatment induced cell death at concentrations as low as 6.25 µM, where QC and LUT treatment resulted in only marginal effects (Figure 2A). The potent stemotoxic effect of LUT was validated in hESCs by PARP-1 and active Caspase 3- and 9-specific immunoblotting. Similar to QC [9], LUT induced mitochondria-mediated cell death, as demonstrated by the formation of active caspases 9 and 3, even at a 10 µM concentration (Figure 2B). Similarly, 12.5 µM of LUT induced evident morphology changes in hESCs (Figure 2C). A robust comparison between LUT and QC would yield important insights, as only QC has been identified as a stemotoxic flavonoid [9] with the capacity to eliminate undifferentiated hPSCs in clinical contexts [3]. Therefore, the potency of LUT was compared to that of QC at low concentrations from 3.125 µM to 50 µM. Dose-response experiments revealed that LUT was more potent than QC (Figure 2D). The IC50 values of LUT and QC, based on the result in Figure 2D, were 9.492 and 16.28 µM, respectively (Appendix A). This result was validated again via single-dose live monitoring of hESCs [Figure 2E and Appendix A] and flow cytometry of active caspase 3 positive population (Figure 2F), as well as caspase 3 activity assays (Figure 2G). Cell death of hESCs after treatment of LUT was significantly reduced in TP53KO hiPSCs compared to WT determined by cell morphological change (Figure 2H) and analysis of flow cytometry for Annexin V/7AAD staining (Figure 2I). These data suggest that the stemotoxic effect of LUT occurred in a p53-dependent manner similar to that of QC [9,16].

### 3.3. Selectivity of Luteolin toward hPSCs

To be used as a stemotoxic agent, candidate compounds must not only effectively induce undifferentiated hPSC death, but also have minimal or no effects on hPSC-derived differentiated cells, as is the case of QC [7]. To examine the safety of LUT to hPSC-derived differentiated cells, we took advantage of isogenic pairs of undifferentiated hiPSCs and smooth muscle cells (SMCs) derived from hiPSCs [21]. Similar to QC [9], the effect of LUT on SMCs appeared minimal, whereas drastic cell death induction was observed in hiPSCs (Figure 3A). The efficacy of LUT to iPSCs and the safety of LUT to SMCs was validated by immunoblotting for PARP-1 cleavage and active caspase 3 and 9, all of which are typical apoptotic responses (Figure 3B). Notably, apoptotic response was manifested in iPSCs expressing sex determining region Y box 2 (SOX2) but not SMCs (Figure 3B). To further confirm the efficacy and safety of LUT, another isogenic pair of undifferentiated hESCs and perivascular progenitor cells (PVPC) derived from hESCs [23] was tested (Figure 3C). To avoid the unexpected bias of the different culture conditions between hESCs and PVPCs, the selectivity of LUT toward undifferentiated hESCs was examined in a mixture of undifferentiated hESCs and PVPCs to imitate the presence of residual undifferentiated hPSCs in exaggeration, as described previously [14]. Shown in Figure 3D, reactivity to active caspase 3 antibody was evident at the sides where hESCs were present, unlike the middle of culturing PVPCs (Figure 3D). Similarly, enhanced green fluorescent protein (EGFP) expressing hESCs (EGFP-hESCs), when co-cultured with PVPCs, significantly disappeared in a dose-dependent manner (Figure 3E). Flow cytometry to trace EGFP-hESCs clearly indicated that LUT treatment eliminated only hESCs but not co-culturing PVPCs, according to no alteration in EGFP negative cells (e.g., PVPCs) (Figure 3F).

### 3.4. Normal Functioning of hESC-Derived Cardiomyocytes after Luteolin Treatment

To examine the effect of LUT on the functionality of hPSC-derived differentiated cells, we took advantage of cardiomyocytes derived from hESCs (hESC-CMs), as their functionality can be readily determined [32,33]. hESC-CMs were differentiated as described in a previous study [24] (Figure 4A). The positive staining of CD172α/β, also known as signal-regulator protein α/β (SIRPα/β), which was previously reported to be a specific surface marker of hESC-CMs [34], was dramatically increased in hESC-CMs, unlike the undifferentiated hESCs (a negative control) (Figure 4B). After differentiation, hESC-CMs exhibited a significant decrease in *POU5F1*, *SOX2* and *NANOG*. (i.e., pluripotency markers) and an increase in *MYH7*, *NKX2-5* and *TNNT2* (i.e., typical cardiomyocyte (CM) markers [35]) compared to undifferentiated hESCs (Figure 4C). After cardiomyocyte differentiation, the cardiomyocytes were further purified with a glucose-free culture medium containing abundant lactate [36]. After purification, a drastic increase in CD172α/β-positive cells was observed (Figure 4D). Similar to that of the other differentiated cells from hPSCs after LUT treatment leading to clear cell death of hPSCs, no clear apoptotic event in hESC-CMs was found (Figure 4E). Next, we aimed to examine any negative effect of LUT to functionality in hESC-CM. The typical functionality of hESC-CMs was observed as spontaneous cell contractions (Figure 4F and Appendix A). The spontaneous beating behavior was readily monitored by the oscillation of intracellular Ca2+ with a Ca2+-specific fluorescence probe, as described in a previous study [37] (Figure 4G and Appendix A). Notably, this procedure has been used in previous studies for in vitro cardiotoxicity assessment of small molecules [38,39]. As expected, no apparent alteration of intracellular Ca2+ oscillation in hPSC-CMs was observed after LUT treatment (Figure 4H and Appendix A), suggesting that LUT treatment would be safe for hESC-CMs while being efficient to eliminate the residual undifferentiated hESCs.

## 4. Discussion

Concerning successful hPSC-based cell therapy, the functionality of the differentiated cells should remain intact during the isolation or maintenance procedures prior to patient transplantation. Thus, it is critically important to confirm whether the functionality of the differentiated cells is unaltered after any approach to selectively eliminate undifferentiated hPSCs [7]. Although a variety of stemotoxic small molecules such as YM155, a potent stemotoxic compound [9], of which the high effectiveness of teratoma inhibition [40,41] are examined, it would be still important to search for new stemotoxic small molecules to ensure the efficacy (toward undifferentiated hESCs and safety (toward differentiated cells) considering the diversity of cells for hPSC-based cell therapy [7]. Moreover, natural flavonoids are known to be safe to normal cells [42] but some, like QC, possess stemotoxic effects [9], thereby providing advantages over other small molecules whose effects on normal cells have not been fully determined. QC treatment, for example, was found to not alter hPSC-derived dopaminergic neuron functionality [9] and, therefore, was used to prevent teratoma formation during iPSC-based clinical trials in Parkinson’s patients [3].

Along with dopaminergic progenitors, hPSC-derived CMs have been approved for heart failure clinical trials [43] and were recently applied to patients in China [44]. Despite many attempts to develop exclusive CM surface markers [34,45], the technical limitations of isolating hPSC-CMs persist. Therefore, undifferentiated hPSCs must be fully eliminated prior to transplantation to the patients. Considering this, LUT is a promising therapeutic agent in heart failure patients, as it was found to not affect hPSC-CM functions (Figure 4) and was more potent against undifferentiated hPSCs than QC (Figure 3). Notably, LUT itself has been shown to be beneficial to cardiomyocytes in vitro and in vivo [46,47]. Therefore, LUT treatment at the end of CM differentiation from hPSCs would not only provide stemotoxic effects but also cardio-protective properties.

Natural flavonoids induce different effects on cancer cells depending on their structure [48]. Therefore, it was interesting that KP and MYC, flavonol compounds with one and three hydroxyl groups at the B ring, exhibited no stemotoxic effects unlike QC, which has two hydroxyl groups (Figure 1A,B). Instead, LUT, a flavone with two hydroxyl groups at the B ring, was found to possess a more potent stemotoxic effect than QC (Figure 2). Considering the significance of a hydrogen bond acceptor in the pyrazine structure of YM155 [9], the stemotoxic effects of LUT may have been due to its interaction to the target protein (possibly SLC35F2) [15,19]. LUT, a flavone with two hydroxyl groups, likely possesses an optimal structure to form hydrogen bonds to putative target proteins and, therefore, its interaction with the target proteins triggers its stemotoxic effects. Nonetheless, the target protein or proteins with which QC and LUT interact to exert their stemotoxic effects have not been identified yet and, therefore, future studies should focus on identifying these targets.

Taken together, our results demonstrated that LUT could serve as a more potent stemotoxic flavonoid than QC. Given that it exhibits an undifferentiated hPSC selectivity similar to that of QC, LUT treatment at the end of differentiation may ensure the complete elimination of undifferentiated hPSCs cells (particularly CMs) while preserving the functions of their differentiated counterparts.

## Figures and Tables

**Figure 1 biomedicines-08-00453-f001:**
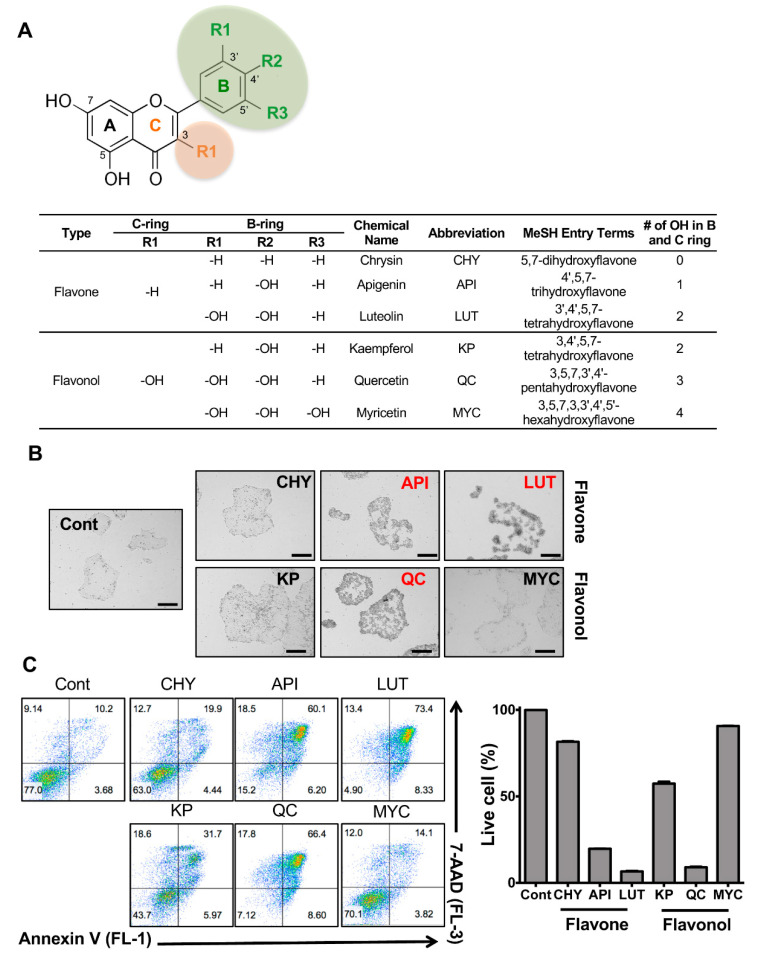
Stemotoxic screening of flavonoids. (**A**) Chemical structure of flavonoids (top) and table (bottom) used in this study (**B**) Microscope images of hESCs 24 h after treatment with 50 µM of indicated flavonoids (scale bar = 200 µm), Flavonoids inducing cell death were indicated in red. (**C**) Flow cytometry for Annexin V/7-AAD assay (left) and graphical presentation of live cells (Annexin V and 7-AAD negative population, right) at 24 h after a 50 µM treatment of indicated flavonoids (*n* = 3).

**Figure 2 biomedicines-08-00453-f002:**
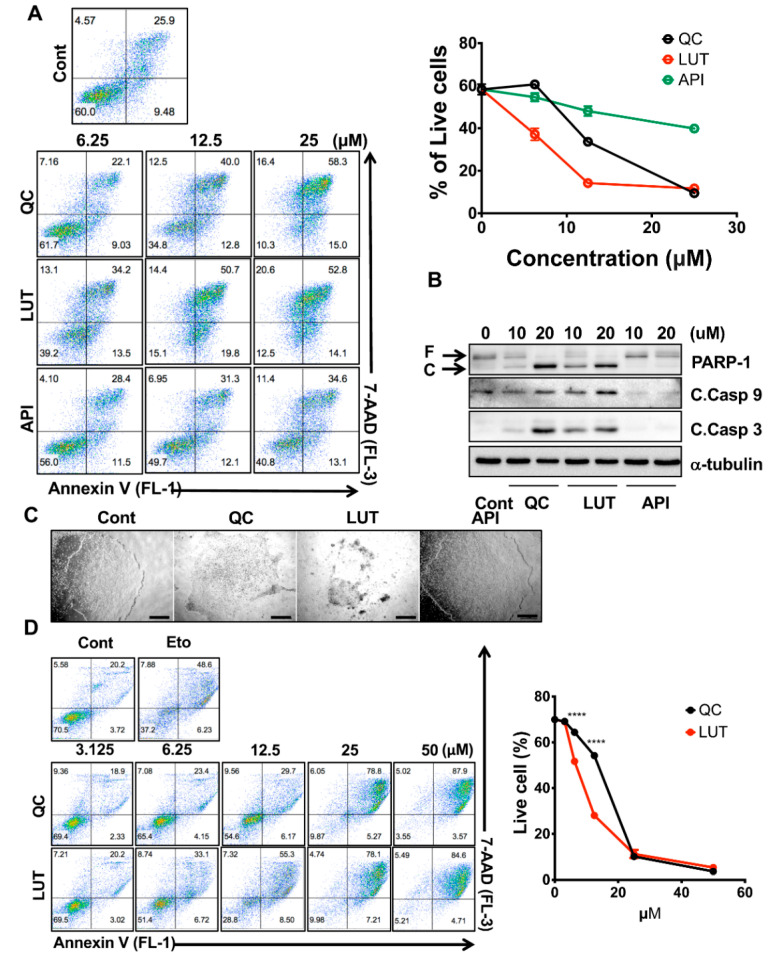
Potent stemotoxic effect of luteolin. (**A**) Flow cytometry for the Annexin V/7-AAD assay (left) and graphical presentation of live cells (Annexin V and 7-AAD negative population, right) 24 h after treatment with indicative concentrations of quercetin (QC), luteolin (LUT) or apigenin (API) (*n* = 2). (**B**) Immunoblotting analysis for PARP-1 (F: full and C: cleaved), cleaved caspase 3 (C.Casp3), cleaved caspase 9 (C.Casp9) at 24 h after indicative concentration of each flavonoid, α-tubulin for equal protein loading. (**C**) Microscope images of hESCs at 24 h after treatment of 12.5 µM of each flavonoid (scale bar = 200 µm). (**D**) Flow cytometry for Annexin V/7-AAD assay (left) and graphical presentation of live cells (Annexin V and 7-AAD negative population, right) at 24 h after indicative concentration of each flavonoid. Treatment of etoposide (Eto) was shown as a positive control of cell death (*n* = 2). *p* < 0.0001 (****). (**E**) Microscope images of hESCs at 48 h after treatment with 12.5 µM of each flavonoid (scale bar = 200 µm). (**F**) Flow cytometry analysis for cleaved caspase 3 (left) and graphical presentation of cleaved caspase 3 positive population after treatment with 12.5 µM of each flavonoid (*n* = 2). (**G**) Graphical presentation of caspase 3 activity at 24 h after treatment of 12.5 µM of each flavonoid (*n* = 2). *p* < 0.05 (*), *p* < 0.01 (**), *p* < 0.001 (***), *p* < 0.0001 (****). (**H** and **I**) Microscope images (scale bar = 200 µm), (**H**) and flow cytometry (**I**) of hiPSCs (WT: wild-type and TP53KO: p53−/− hiPSCs) at 24 h after treatment of 12.5 µM of luteolin (LUT), graphical presentation of live cells was shown (right) (*n* = 2).

**Figure 3 biomedicines-08-00453-f003:**
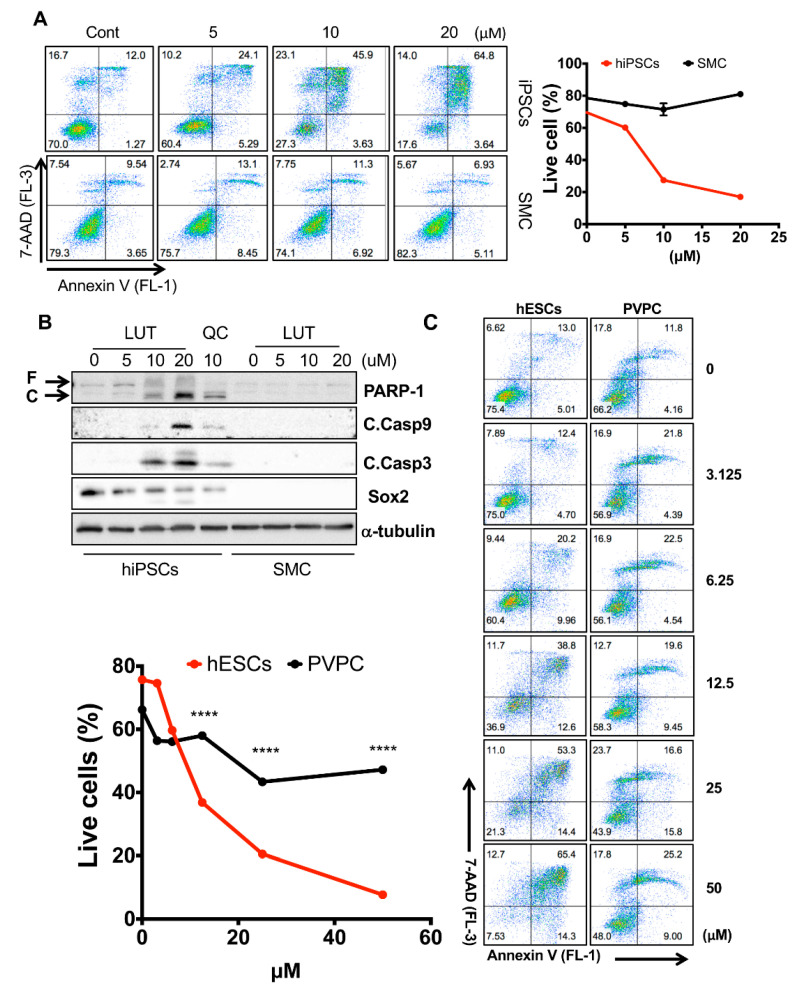
Selectivity of luteolin toward hPSCs. (**A**) Flow cytometry for Annexin V/7-AAD assay (left) and graphical presentation of live cells (Annexin V and 7-AAD negative population, right) 24 h after treatment of an indicative concentration of luteolin in hiPSCs and SMC (*n* = 2). (**B**) Immunoblotting analysis for PARP-1 (F: full and C: cleaved), cleaved caspase 3 (C.Casp3), cleaved caspase 9 (C.Casp9) in hiPSCs and SMC at 24 h after treatment with the indicated concentrations of each flavonoid, and α-tubulin for equal loading control. (**C**) Flow cytometry for Annexin V/7-AAD assay (right) and graphical presentation of live cells (Annexin V and 7-AAD negative population, left) 24 h after treatment with the indicative concentrations of luteolin in hESCs and PVPCs (*n* = 2). *p* < 0.0001 (****). (**D**) Immunofluorescent microscope images of a mixture of hESC and PVPCs at 24 h after LUT treatment (12.5 µM). White spotted lines divide hESCs and PVPCs (scale bar = 50 µm). (**E**) Microscope images (top) and fluorescent images (bottom) of a mixture of EGFP expressing hESCs and PVPCs at 24 h after indicated the concentration of luteolin (LUT) (scale bar = 200 µm). (**F**) Flow cytometry (left) and graphical presentation (right) for the GFP positive population in the mixture of EGFP-hESCs and PVPC at 24 h after the indicative concentration of luteolin (LUT) treatment (*n* = 2). *p* < 0.001 (***), *p* < 0.0001 (****).

**Figure 4 biomedicines-08-00453-f004:**
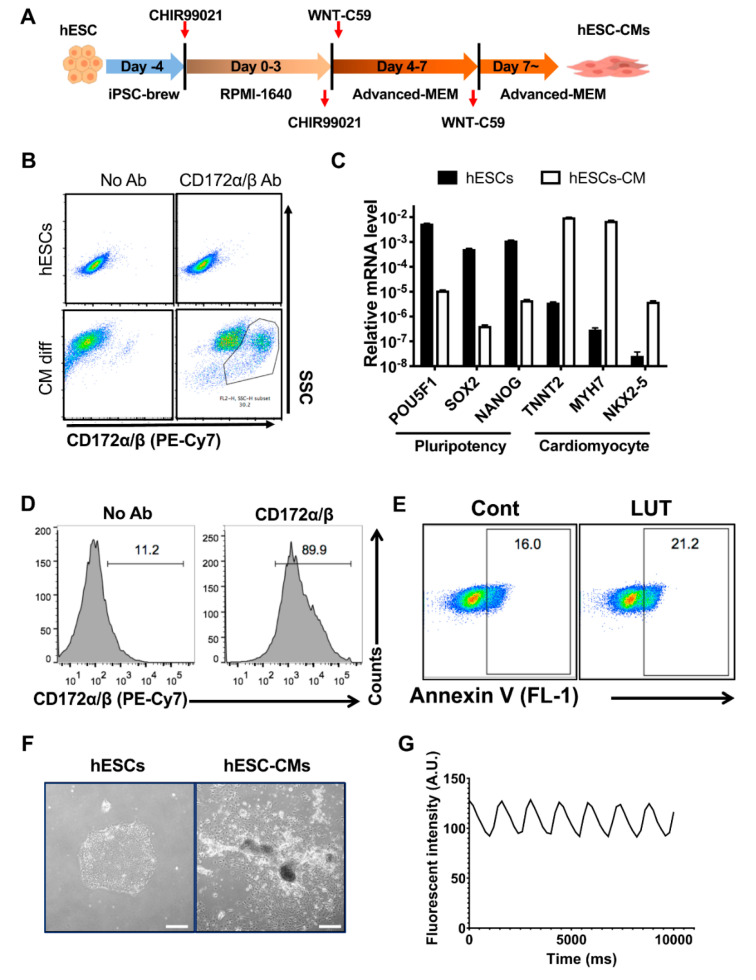
Normal functioning of hESC-derived cardiomyocytes after luteolin treatment. (**A**) Schematic diagram for procedure of cardiomyocyte differentiation. (**B**) Flow cytometry for CD172α/β in hESCs or hESC-CMs after cardiomyocyte differentiation (No Ab: negative control without antibody, CD172α/β Ab: CD172α/β antibody) (*n* = 2). (**C**) Graphical presentation of relative expression levels of pluripotency (*POU5F1, SOX2, NANOG*) and cardiomyocyte (*TNNT2, MYH7, NKX2-5*) markers of hESCs and hESC-CMs (*n* = 2). (**D**) Flow cytometry for CD172α/β of hESC-CMs (No Ab: negative control without antibody, CD172α/β Ab: CD172α/β antibody). (**E**) Flow cytometry for Annexin V in hESC-CMs at 24 h after luteolin (LUT) treatment (12.5 µM). (**F**) Microscope image of undifferentiated hESCs (left) and hESC-CMs (right) (scale bar = 200 µm). (**G**,**H**) Graphical presentation of mean intensity of fluorescence of Ca^2+^ in hESC-CMs; (**G**), with 24 h treatment of mock (**H**: left) or 12.5 µM of LUT (**H**: right) (*n* > 70).

**Table 1 biomedicines-08-00453-t001:** Primer Sequences for real-time PCR analysis.

Gene	Forward Sequence (5′ to 3′)	Reverse Sequence (5′ to 3′)
*POU5F1*	GTGGAGGAAGCTGACAACAA	ATTCTCCAGGTTGCCTCTCA
*SOX2*	TTCACATGTCCCAGCACTACCAGA	TCACATGTGTGAGAGGGGCAGTGTGC
*NANOG*	AAATTGGTGATGAAGATGTATTCG	GCAAAACAGAGCCAAAAACG
*TNNT2*	ATGAGCGGGAGAAGGAGCGGCAGAAC	TCAATGGCCAGCACCTTCCTCCTCTC
*MYH7*	CACCAACAACCCCTACGATT	ACTCATTGCCCACTTTCACC
*NKX* *2-5*	GTTCCAGAACCGGCGCTACAAGTG	GCTTGCCATCGCGCACCAGCACTG

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
