# Peer review of "Luteolin Induces Selective Cell Death of Human Pluripotent Stem Cells"

_biomedicines, 2020, doi:10.3390/biomedicines8110453_

Round 1
Reviewer 1 Report
In this study Young-Hyun Go and collaborators demonstrate that Luteolin is a promising approach to minimize teratoma formation risk, more potent than Quercetin.
Overall the paper is interesting and data convincing. However, the following issues should be addressed in order to improve this study for the readers of Biomedicines:
- Authors should make the results clear and better organized.
- Figure 1B: Please explain the white boxes. What they indicate? I would suggest to remove them.
- Line 167: Please clarify “the results of the preliminary experiments are illustrated in Figure 1” as the results of the dose dependent effects of three flavonoids are not shown in figure 1
- Line 177: “IC50 values of LUT and QC were 9.492 and 16.28 uM respectively”. Where are these data shown?
- Please better check the description of Figure 2 results. In particular, describe Fig 2D in the result section.
- Results of figure 2H and 2I are missed. Please add results description.
- Please better check the entire figure 2 legend. Description of Figure 2H and 2I are missed.
- Authors provide data of selectivity of LUT in a mixture of undifferentiated hESCs and PVPCs. Please provide information regarding the methods used.
- Results of Figure 3C-D-E-F are not described or are indicated in the results as figure 4 by mistake. Authors should be more clear in the results description, please check and make clear every figure results and legends.
- Figure 4C. Did the authors analyze also other pluripotency markers and markers of cardiomyocyte differentiation? What is the rationale in showing only NANOG as pluripotent marker and TNNT2 as differentiated cell marker? Authors should provide data of other pluripotency markers in pluripotent cells and their expression following cardiomyocyte differentiation.
- It is not clear if human perivascular progenitor cells (PVPCs) derived from hESCs and human smooth muscle cell derived from human iPSCs were purchased or differentiation experiments were performed. Please provide information in material and methods.
- Please clarify differences among movie S1A, S1B and S1C. Same for movies S3A and S3B.
Author Response
Authors should make the results clear and better organized.
Figure 1B: Please explain the white boxes. What they indicate? I would suggest to remove them.
- It was intended to group each flavonoid based on the number of hydroxyl group in B ring. However, as per the reviewer’s comment, the white boxes were removed to avoid the unnecessary confusion.
Line 167: Please clarify “the results of the preliminary experiments are illustrated in Figure 1” as the results of the dose dependent effects of three flavonoids are not shown in figure 1
- The initial screening of stemotoxic flavonoid was performed with a single dose (50mM) shown in Figure 1B and C. LUT, API and QC were selected due to the cytotoxic effect to undifferentiated hESCs (Figs. 1B and C). As per the reviewer’s comment, the sentence was edited as following. “After determining the stemotoxic effect of flavonoids with a single dose (50 mM) (Fig. 1B and C), three flavonoids (API, LUT and QC) were selected for further validation to examine the dose-dependent effects toward undifferentiated hESCs.”
Line 177: “IC50 values of LUT and QC were 9.492 and 16.28 uM respectively”. Where are these data shown?
- IC50 of LUT or QC was obtained based on the data in Figure 2D using GraphPad Prism (v8.0). The IC50 plot from GraphPad Prism was shown in Figure S2.
Please better check the description of Figure 2 results. In particular, describe Fig 2D in the result section.
- As per the reviewer’s comment, the description of Figure 2 was reviewed and edited as following. “For this purpose, the potency of LUT was compared to that of QC at low concentrations from 3.125m to 50m Dose-response experiments revealed that LUT was more potent than QC (Fig. 2D). The IC50 values of LUT and QC, based on the result in Figure 2D, were 9.492 and 16.28 mM respectively (Fig. S2).”
Results of figure 2H and 2I are missed. Please add results description.
- Results of figure 2H and 2I were added to the manuscript just prior to the submission so that the description in the main text was accidently omitted. The results description for figure 2H and 2I was added in the main text as following. “Cell death of hESCs after treatment of LUT was significantly reduced in TP53KO hiPSCs compared to WT determined by cell morphological change (Fig. 2H) and analysis of flow cytometry for Annexin V/7AAD staining (Fig. 2I). These data suggest that stemotoxic effect of LUT occurred in p53 dependent manner as similar as that of QC.”
Please better check the entire figure 2 legend. Description of Figure 2H and 2I are missed.
- It is our apology. Figure legends for Figure 2H and I were added.
Authors provide data of selectivity of LUT in a mixture of undifferentiated hESCs and PVPCs. Please provide information regarding the methods used.
- To examine the selective cell death of hESCs by LUT, the fluorescence based competition assay was performed as previously described [1]. The brief method was described in the Method session.
Results of Figure 3C-D-E-F are not described or are indicated in the results as figure 4 by mistake. Authors should be more clear in the results description, please check and make clear every figure results and legends.
- It is our apology for unnecessary confusion. We double-checked the description and edited according to the figures. The main text was carefully reviewed and edited accordingly. As shown in Figure 3D, reactivity to active caspase 3 antibody was evident at the sides where hESCs were present unlike the middle of culturing PVPCs (Fig. 3D). In similar, enhanced green fluorescent protein (EGFP) expressing hESCs (EGFP-hESCs) when co-cultured with PVPCs significantly disappeared in dose dependent manner (Fig. 3E). Flow cytometry to trace EGFP-hESCs clearly indicated that LUT treatment eliminated only hESCs but not co-culturing PVPCs according to no alteration in EGFP negative cells (e.g. PVPCs) (Fig. 3F).
Figure 4C. Did the authors analyze also other pluripotency markers and markers of cardiomyocyte differentiation? What is the rationale in showing only NANOG as pluripotent marker and TNNT2 as differentiated cell marker? Authors should provide data of other pluripotency markers in pluripotent cells and their expression following cardiomyocyte differentiation.
- As per the reviewer’s comment, other markers for pluripotency such as POU5F1 and SOX2 and cardiomyocyte differentiation such as MYH7 and NKX2-5 were determined. This result may support that cardiomyocyte differentiation was valid (Fig. 4C).
It is not clear if human perivascular progenitor cells (PVPCs) derived from hESCs and human smooth muscle cell derived from human iPSCs were purchased or differentiation experiments were performed. Please provide information in material and methods.
- We cited the papers to firstly report these cells in the Material session. As per the reviewer’s comment, information of these cells was added in the Material and methods session.
Please clarify differences among movie S1A, S1B and S1C. Same for movies S3A and S3B.
- The information was added in the supplement figure legends. For clarification, these information was provided in the main text.
Reviewer 2 Report
Go et al. described the effect of luteolin on cell death of hPSC in this manuscript. However, this manuscript has several critical problems and should be carefully revising throughout.
For examples
- On page 3, line 134, authors described” Therefore, hPSCs express higher levels of SLC2A1 (Solute Carrier Family 2 Member 1), encoding GLUT-1, a glucose transporter protein than human dermal fibroblasts (hDFs) (Fig. 1A).” On the other hand, fig1A shows chemical structure of flavonoids. Sentence is not corresponding to figure.
- The sentences about SLC2A1 and QC-glucoside confuse readers. Do authors think that SLC2A1 is key protein for stemotoxic effect of falavonids including QC and LUT?
- On page 4, line 157, authors described” As similar as QC-glycoside, glycoside of KP and LUT gave negligible effect on hPSCs (Fig. 1B)”. However, result of QC-glycoside is not indicated in Fig1B.
- Why blank squares are showed in Fig1B?
- Authors should add the number of samples or assay replications in all data.
- Authors should describe the results showing Fig2H and I in the manuscript.
- Authors should add flowcytometric analysis of CD172 alpha/beta in undifferentiated hESC in Fig4B.
- Whereas authors showed the relative expression levels of NANOG and TNNT21 in fig4C, detail procedure is not indicated in materials and methods. Please provide details.
Author Response
On page 3, line 134, authors described” Therefore, hPSCs express higher levels of SLC2A1 (Solute Carrier Family 2 Member 1), encoding GLUT-1, a glucose transporter protein than human dermal fibroblasts (hDFs) (Fig. 1A).” On the other hand, fig1A shows chemical structure of flavonoids. Sentence is not corresponding to figure.
- It is our apology for the mistake. The sentence is the description for Figure S1A. The typo was corrected.
The sentences about SLC2A1 and QC-glucoside confuse readers. Do authors think that SLC2A1 is key protein for stemotoxic effect of flavonoids including QC and LUT?
On page 4, line 157, authors described” As similar as QC-glycoside, glycoside of KP and LUT gave negligible effect on hPSCs (Fig. 1B)”. However, result of QC-glycoside is not indicated in Fig1B.
- Considering the high expression of GLUT-1 in hESCs (Fig. S1A), we first aimed to potentiate QC induced stemotoxic effect to undifferentiated hESCs using QC-glycoside. This strategy is in similar with the cancer targeting approach using the glucose conjugation to anti-cancer drugs on the basis of high GLUT-1 expression in cancer cells [2]. However, unexpectedly, no stemotoxic effect was found in QC-glycoside. This result was shown in Figure S1B. As per the reviewer’s comment, the main text was corrected.
Why blank squares are showed in Fig1B?
- It was intended to group each flavonoid based on the number of hydroxyl group in B ring. However, as per the reviewer’s comment, the white boxes were removed to avoid the confusion.
Authors should add the number of samples or assay replications in all data.
- As per the reviewer’s comment, the number of samples or assay replications were added in the figure legends.
Authors should describe the results showing Fig2H and I in the manuscript.
- Results of figure 2H and 2I were added to the manuscript just prior to the submission so that the description in the main text was accidently omitted. The results description for figure 2H and 2I was added in the main text as following. “Cell death of hESCs after treatment of LUT was significantly reduced in TP53KO hiPSCs compared to WT determined by cell morphological change (Fig. 2H) and analysis of flow cytometry for Annexin V/7AAD staining (Fig. 2I). These data suggest that stemotoxic effect of LUT occurred in p53 dependent manner as similar as that of QC.”
Authors should add flowcytometric analysis of CD172 alpha/beta in undifferentiated hESC in Fig4B.
- As per the reviewer’s comment, flow cytometry data of hESCs with CD172a/b was added in the Figure 4B. As expected, no positive population for CD172a/b was found in undifferentiated hESCs. The description of Fig. 4B was added in result and legend session.
Whereas authors showed the relative expression levels of NANOG and TNNT21 in fig4C, detail procedure is not indicated in materials and methods. Please provide details.
- Thank you for the comment from the reviewer. We described the detail protocol for quantitative real-time PCR in the Materials session.
Reviewer 3 Report
The manuscript by Young-Hyun Go et al. aimed to identify flavonoids which are able to selectively induce the death of human pluripotent stem cells, thus avoiding the potential formation of teratoma due to presence of undifferentiated stem cells.
Towards this goal they screened 6 in-house flavonoids, identifying luteolin as able to induce hPSCs apoptosis by activation of caspase 3. Moreover, they demonstrated that luteolin determined a selective apoptosis of human embryonic stem cells (hESC) with only marginal effects on hESC-derived cardiomyocytes. The manuscript is well-designed and written. Topic and methodology are sound. Conclusion are supported by the results.
I have the following comments:
- Results section (pages 3-4 lines 132-144): the authors reported the possibility to conjugate glucose to drug molecules to improve their delivery. This was already described in ref 9, 26-29, cited by the authors. Please move this section in the introduction or discussion and focus on new findings in the results section.
- Figure 1 and results section (pages 4-5 lines 147-163): As initial screening, the authors assessed the stemotoxic effect of flavonoids with a single dose of 50 uM. Please consider to evaluate a dose-response effect for all di screened flavonoid (maybe a flavonoid could be toxic at higher concentration than another) and/or report a reference supporting the dose used in this study.
- Results section (page 5 lines 165-175): Dose dependent effect was evaluated for 3 of 6 flavonoids; please consider including other flavonoids considering the dose-response effect of the other 3 (see my previous point);
- Results section (pages 7-8 lines 197-214): the selectivity towards hPSC was evaluated only for luteolin. The excluded compounds could be less toxic but more selective; please discus;
- Luteolin is more potent than quercetin, as demonstrated by the authors. But there was really the need for another drug with the same properties and purpose? Please highlight elements of novelty and discuss other potential advantages and disadvantages of luteolin compared to other flavonoids and to quercetin.
Author Response
Results section (pages 3-4 lines 132-144): the authors reported the possibility to conjugate glucose to drug molecules to improve their delivery. This was already described in ref 9, 26-29, cited by the authors. Please move this section in the introduction or discussion and focus on new findings in the results section.
- With full respect to the reviewer, the result description for QC-glycoside would be necessary to stay in the result session because the data (despite negative result) were presented in Figure S1B. Unlike the comment from the reviewer, the effect of QC-glycoside had not tested in the Ref#9 (Lee MO et. al. PNAS 2013). The papers cited in the manuscript (Ref#29 - #31) examined the strategy using glucose conjugated drugs in cancer cells.
Figure 1 and results section (pages 4-5 lines 147-163): As initial screening, the authors assessed the stemotoxic effect of flavonoids with a single dose of 50 uM. Please consider to evaluate a dose-response effect for all di screened flavonoid (maybe a flavonoid could be toxic at higher concentration than another) and/or report a reference supporting the dose used in this study. Results section (page 5 lines 165-175): Dose dependent effect was evaluated for 3 of 6 flavonoids; please consider including other flavonoids considering the dose-response effect of the other 3 (see my previous point);
- As QC was identified as the stemotoxic flavonoid [3], of which safety toward the differentiated cells was validated [4], we aimed to search for more potent stemotoxic flavonoid. In this line, as similar as the previous study to screen the derivatives of YM155 (another stemotoxic small molecule [3]) for more potent stemotoxic compound [5], the initial screening was made with a single dose. As per the reviewer’s comment, it is still possible that higher concentration of flavonoids would induce cytotoxicity in hPSCs. However, this was far from the aim of this study. We also recognized that flavones (e.g. API and LUT) was more potent than flavonols (e.g. QC) with same number of hydroxyl group at B ring with unknown reasons. This was briefly described in the discussion. Thus, it would be less relevant to check the dose dependent effect of the other flavonoids with no stemotoxic effect.
Results section (pages 7-8 lines 197-214): the selectivity towards hPSC was evaluated only for luteolin. The excluded compounds could be less toxic but more selective; please discus;
- We appreciate the reviewer’s comment. As per the reviewer’s comment, this possibility was briefly described in the discussion as following.
Luteolin is more potent than quercetin, as demonstrated by the authors. But there was really the need for another drug with the same properties and purpose? Please highlight elements of novelty and discuss other potential advantages and disadvantages of luteolin compared to other flavonoids and to quercetin.
- We agree to the reviewer’s critical point. QC was used in the first clinical trial of iPSC based Parkinson’s therapy [4] due to the functional safety of QC to dopaminergic neurons [3] as well as the long-term safety validation as a dietary supplement [6]. In addition, we had identified another potent stemotoxic small molecule, YM155, working in the nanomole range [3]. The efficacy of YM155 to inhibit teratoma formation was validated in the independent studies [7, 8]. According to the recent study from us, YM155 selectivity is determined by expression level of SLC35F2, a solute carrier protein [9]. Thus, the safety of differentiated cells from hPSCs by YM155 treatment would be readily validated by examining SLC35F2 expression in the target cells. In general, functional safety of the differentiated cells by stemotoxic small molecules would be the primary requirement as well as the efficacy to the differentiated cells [10]. In this regard, considering diversity of the differentiated cells derived from hPSCs for future cell therapy, continuous identification of stemotoxic small molecules of which its functional safety to the differentiated cells is verified, would have its own significance. In this study, we showed the evidence that cardiomyocyte derived from hESCs, remained fully functional after LUT treatment. Considering the potency of LUT to hPSCs, LUT would be an important option for teratoma-free engraft of cardiomyocytes from hPSCs, which is currently under clinical trials [11]. This was briefly described in the discussion session.

Round 2
Reviewer 2 Report
The manuscript have been revised by authors well.
Reviewer 3 Report
Thank you to the authors for their kind responses. All points have been answered to my satisfaction, and I recommend the manuscript for publication.